# Clinical Impact of the Use of Warfarin in Patients with Atrial Fibrillation Undergoing Maintenance Hemodialysis

**DOI:** 10.3390/jcm13082404

**Published:** 2024-04-20

**Authors:** Seok Hui Kang, Gui Ok Kim, Bo Yeon Kim, Eun Jung Son, Jun Young Do

**Affiliations:** 1Division of Nephrology, Department of Internal Medicine, College of Medicine, Yeungnam University, Daegu 42415, Republic of Korea; kangkang@ynu.ac.kr; 2Healthcare Review and Assessment Committee, Health Insurance Review and Assessment Service, Wonju 26465, Republic of Korea; 3Quality Assessment Department, Health Insurance Review and Assessment Service, Wonju 26465, Republic of Korea

**Keywords:** atrial fibrillation, hemodialysis, warfarin, survival

## Abstract

**Background**: We evaluated the impact of warfarin use on the clinical outcomes of patients with atrial fibrillation who were undergoing hemodialysis (HD). **Methods**: A retrospective analysis was conducted utilizing data from patients undergoing maintenance HD who participated in HD quality assessment programs. Patients who were assigned the diagnostic code for atrial fibrillation (n = 4829) were included and divided into two groups based on the use of warfarin: No group (no warfarin prescriptions (n = 4009)), and Warfarin group (warfarin prescriptions (n = 820)). **Results**: Cox regression analyses revealed that the hazard ratio for all-cause mortality in the Warfarin group was 1.15 (*p* = 0.005) in univariate analysis and 1.11 (*p* = 0.047) in multivariable analysis compared to that of the No group. Hemorrhagic stroke was significantly associated with warfarin use, but no significant association between the use of warfarin and ischemic stroke or cardiovascular events was observed. The subgroup results demonstrated similar trends. **Conclusions**: Warfarin use is associated with a higher risk of all-cause mortality and hemorrhagic stroke, and has a neutral effect on ischemic stroke and cardiovascular events in patients with atrial fibrillation who are undergoing HD, compared to those who are not using warfarin.

## 1. Introduction

The incidence of chronic kidney disease, a condition that can progress to end-stage kidney disease (ESKD) and require renal replacement therapy, has rapidly increased. Hemodialysis (HD) is the most widely used dialysis modality globally; considering the high mortality rate of patients on dialysis, improving their survival is crucial [1,2]. Cardiovascular disease is a prevalent comorbidity in patients undergoing HD; atrial fibrillation is common and contributes to complications and elevated mortality rates. Traditionally, atrial fibrillation is associated with a high thromboembolic risk, leading to the use of anticoagulants such as warfarin [3]. However, in patients undergoing HD, impaired platelet function and the use of medications such as heparin contribute to a heightened bleeding diathesis [4,5]. Anticoagulation in such patients may, paradoxically, result in increased complication rates or mortality [6]. In contrast, patients undergoing HD are prone to hypercoagulability, which can be associated with increased thrombotic complications [7,8]. Therefore, warfarin should be used carefully in patients undergoing HD.

Guidelines from the European Society of Cardiology and the American Heart Association recommend anticoagulant usage in patients with atrial fibrillation and CHA_2_DS_2_-VAS_C_ scores ≥ 1 in males and ≥2 in females [9,10]. Therefore, most patients undergoing HD would require anticoagulant therapy. It is beneficial to administer anticoagulants based on these criteria in patients not undergoing dialysis; however, patients undergoing HD do not demonstrate the same outcomes [11,12,13]. A meta-analysis of 15 cohort studies revealed that warfarin administration results in higher rates of hemorrhagic stroke, but has a neutral effect on ischemic stroke and all-cause mortality [6]. However, the studies included in this meta-analysis had relatively short follow-ups (median, 2.6 years) and significant clinical heterogeneity. Therefore, clear conclusions or criteria related to the use of warfarin in patients undergoing HD cannot be drawn based on these factors. In this study, we used the laboratory and clinical findings associated with HD quality of a large cohort to evaluate the impact of warfarin use on the clinical outcomes of patients with atrial fibrillation who were undergoing HD.

## 2. Materials and Methods

### 2.1. Data Source and Study Participants

A retrospective analysis was conducted utilizing data from adult patients (age ≥ 18 years) undergoing maintenance HD (≥3 months and ≥2 times per week) who participated in the fourth (July and December 2013), fifth (July and December 2015), and sixth (March and August 2018) HD quality assessment programs performed by the Health Insurance Review and Assessment (HIRA) of the Republic of Korea [14]. In total, 21,846, 35,538, and 31,294 patients were included in the fourth, fifth, and sixth HD quality assessment programs, respectively. We excluded repetitive participants (75 participants were from the 4th, 13,795 from the 5th, and 18,570 from the 6th programs), patients with insufficient data (n = 19), or those who had undergone HD with a catheter (n = 1316) (Appendix A).

Among 54,903 patients, we included 4863 who were diagnosed with atrial fibrillation, using ICD-10 code I48, for at least 18 months (6 months during the HD quality assessment and the preceding 12 months). In addition, we excluded patients who had been administered anticoagulants other than warfarin (n = 5), those who had used warfarin for <30 days, or those who had been administered ≥ 2 anticoagulants (n = 29) during the 6 months of the HD quality assessment program. Finally, we included 4829 patients. The institutional review board of Yeungnam University Medical Center approved this study (approval no. YUMC 2024-02-013). The need for informed consent was waived because the records and information of patients were anonymized and de-identified before the analysis. All methods were performed in accordance with the Declaration of Helsinki and relevant guidelines/regulations.

### 2.2. Variables

We collected data on age, sex, body mass index (BMI) (kg/m^2^), HD vintage (months), the presence of diabetes as an underlying cause of ESKD, and vascular access types. Hemoglobin levels (g/dL), Kt/V_urea_, serum albumin concentrations (g/dL), serum calcium levels (mg/dL), serum phosphorus levels (mg/dL), serum creatinine levels (mg/dL), and ultrafiltration volumes (UFV) (L/session) were recorded. Information was gathered monthly, with laboratory values averaged monthly. The Kt/V_urea_ was determined using the Daugirdas equation [15].

The codes corresponding to medications can be found in Appendix A. Patients were divided into two groups based on the use of warfarin: No group (no warfarin prescriptions (n = 4009)), and Warfarin group (warfarin prescriptions for ≥30 days during the 6 months of the HD quality assessment program (n = 820)). We evaluated the use of aspirin, clopidogrel, statins, and β-blockers, which were defined as being prescribed for ≥30 days. Comorbidities were assessed using the Charlson Comorbidity Index (CCI) and evaluated during the year before the HD quality assessment program. CCI scores were computed for all patients using ICD-10 codes, as previously described [16,17,18]. In addition, we evaluated the CHA_2_DS_2_-VAS_C_ and HAS-BLED scores, as previously described [19,20]. Our study did not collect data for alcohol and labile international normalized ratios; therefore, we used a modified HAS-BLED score that excluded these two variables [11]. We defined a low CHA_2_DS_2_-VAS_C_ score as <2 in females and <3 in males, and a high CHA_2_DS_2_-VAS_C_ score as ≥2 in females and ≥3 in males.

### 2.3. Outcomes

Patients were followed up with until April 2022. We evaluated all-cause mortality as the primary outcome, and hemorrhagic stroke, ischemic stroke, and cardiovascular events (CVE) as secondary outcomes. We evaluated the incidence of hemorrhagic stroke (I60, I61, I62) or ischemic stroke (I63) using the relevant ICD-10 codes after the end of the HD quality assessment program, and that of CVE, including myocardial infarction, stroke, and revascularization, regardless of survival or death, as shown in Appendix A [14]. To assess each secondary outcome, we analyzed patients without prior diagnosis of each condition in the year preceding the HD quality assessment period. Data on patient deaths were obtained from the HIRA, and patients whose treatment changed to peritoneal dialysis or who underwent renal transplantation without experiencing an event were censored at the time of transfer.

### 2.4. Statistical Analyses

The data were analyzed using two statistical software programs (SAS Enterprise Guide [version 7.1; SAS Institute, Cary, NC, USA] and R [version 3.5.1; R Foundation for Statistical Computing, Vienna, Austria]). Categorical variables are represented as frequencies and percentages, while continuous variables are represented as means and standard deviations. We used Pearson’s χ^2^ test or Fisher’s exact test to evaluate the statistically significant difference between categorical variables. Differences in continuous variables among the two groups were assessed using the *t*-test.

Survival curves were estimated using Kaplan–Meier curves. *P*-values for comparing survival curves were determined using the log-rank test. Hazard ratios (HRs) and confidence intervals (CIs) were calculated using Cox regression analyses. Age, BMI, vascular access type, sex, HD vintage, CCI score, diabetes, UFV, Kt/V_urea_, and levels of creatinine, albumin, phosphorus, hemoglobin, and calcium were all taken into account in the multivariable Cox regression analyses. Moreover, the analyses accounted for the use of aspirin, clopidogrel, β-blockers, or statins. The multivariable Cox regression analyses utilized the enter mode. In the multivariable analysis, CHA_2_DS_2_-VAS_C_ and HAS-BLED scores were excluded from the set of covariates due to their statistical correlation with other covariates to prevent multicollinearity and improve the stability of the model. 

To balance the baseline characteristics between the No and Warfarin groups, we estimated propensity scores using logistic regression models and the following variables: age, sex, BMI, vascular access type, CCI score, HD vintage, diabetes, UFV, Kt/V_urea_, levels of hemoglobin, albumin, creatinine, phosphorus, and calcium, and the use of aspirin, clopidogrel, β-blockers, or statins. Participants in the Warfarin group were matched with participants in the No group using 1:2 nearest neighbor matching without replacement and with a matching tolerance (caliper) of 0.2; the nearest neighborhood matching was based on propensity scores. Statistical significance was set at *p* < 0.05.

## 3. Results

### 3.1. Participants’ Baseline Characteristics

Baseline characteristics are shown in Table 1.

Sex, presence of diabetes, use of β-blockers, HD vintage, BMI, CCI scores, Kt/V_urea_, UFV, serum albumin concentrations, calcium levels, and CHA_2_DS_2_-VAS_C_ scores were not significantly different between the two groups. The number of patients with arteriovenous fistula in the No and Warfarin groups was 3257 (81.2%) and 666 (81.2%), respectively. However, the Warfarin group had older participants and had greater hemoglobin levels and usage rates of statins than the No group participants. In addition, the Warfarin group participants had higher usage rates of statins and lower serum creatinine and phosphorus levels, HAS-BLED scores, and usage rates of clopidogrel or aspirin than the No group participants.

### 3.2. Survival Analysis

The follow-up durations in the No and Warfarin groups were 53 ± 28 and 50 ± 27 months, respectively. The number of patient deaths at the follow-up end-point was 2178 (54.3%) in the No group and 476 (58.0%) in the Warfarin group (*p* = 0.056). The 5-year patient survival rates in the No and Warfarin groups were 53.7% and 46.7%, respectively (Figure 1A, *p* = 0.005). 

Cox regression analyses revealed that the HR for all-cause mortality in the Warfarin group was 1.15 (95% CI, 1.05–1.28; *p* = 0.005) in univariate analysis and 1.11 (95% CI, 1.01–1.23, *p* = 0.047) in the multivariable analysis compared to that in the No group (Table 2). 

### 3.3. Cerebrovascular Event Analyses

The 5-year hemorrhagic stroke-free survival rates in the No and Warfarin groups were 93.3% and 90.8%, respectively (Figure 1B, *p* = 0.002). The 5-year ischemic stroke-free survival rates in the No and Warfarin groups were 81.0% and 81.3%, respectively (Figure 1C, *p* = 0.850). The 5-year CVE-free survival rates in the No and Warfarin groups were 83.6% and 84.6%, respectively (Figure 1D, *p* = 0.570). Cox regression analyses revealed that hemorrhagic stroke was significantly associated with the use of warfarin, but no statistically significant association between the use of warfarin and ischemic stroke or CVE was observed (Table 2).

### 3.4. Subgroup Analyses

We performed subgroup analyses based on sex, age, presence of diabetes, and the CHA_2_DS_2_-VAS_C_ score. Univariate analyses showed that all-cause mortality was associated with the use of warfarin in females, an age of < 65 years, the absence of diabetes, and high CHA_2_DS_2_-VAS_C_ score subgroups (Figure 2A). 

Hemorrhagic stroke was associated with the use of warfarin in males, two age groups, the presence of diabetes, and high CHA_2_DS_2_-VAS_C_ score subgroups, and CVE was inversely associated with the use of warfarin in females (Figure 2B–D). Multivariable Cox regression analyses revealed that the use of warfarin was positively associated with all-cause mortality in females and patients aged < 65 years (Figure 3A), and with hemorrhagic stroke in all subgroups except females, and was inversely associated with CVE in females and patients without diabetes (Figure 3B–D). 

Ischemic stroke was not associated with the use of warfarin in any subgroup in univariate and multivariable analyses.

### 3.5. Factors Associated with All-Cause Mortality or Hemorrhagic Stroke

In multivariate analysis, all-cause mortality was positively associated with age, presence of diabetes, serum calcium and phosphorus levels, HD vintage, CCI scores, UFV, and the use of clopidogrel, and was inversely associated with females, BMI, Kt/V_urea_, and levels of hemoglobin, albumin, and creatinine (Appendix A). In multivariate analysis, hemorrhagic stroke was positively associated with the presence of diabetes and serum calcium levels but was inversely associated with the BMI. 

### 3.6. Analyses Using Cohort after Propensity Score Matching

We performed analyses using cohort after propensity score matching. The distribution of propensity scores after matching were similar between the two groups compared to that before matching (Appendix A). Differences in most baseline characteristics were attenuated after matching (Appendix A). Cox regression analyses using cohort after propensity score matching were similar to those using total cohort (Appendix A).

## 4. Discussion

We analyzed clinical, laboratory, and claims data of patients who underwent maintenance HD as part of the HD quality assessment program. We included patients with diagnostic codes for atrial fibrillation and divided them into two groups based on the use of warfarin. The number of patients receiving anticoagulation therapy in addition to warfarin was small, and they were excluded from our study. Our study revealed that the use of warfarin was associated with an increased risk of all-cause mortality and hemorrhagic stroke. However, ischemic stroke and CVE were not related to the use of warfarin in our study, and similar trends were observed in subgroup analyses. In addition, we evaluated independent risk factors for hemorrhagic stroke; the results showed that the risk of hemorrhagic stroke increased as diabetes risk and serum calcium levels increased, whereas it decreased as the BMI increased.

Patients undergoing HD are prone to bleeding diathesis, which is associated with various factors. Thrombocytopenia can develop due to varied medications or procedures, including heparin and blood-hemodialyzer contact [4]. Platelet dysfunction, including structural or functional abnormalities, is a common complication in patients undergoing HD. Structural changes (e.g., decreased mean platelet volume), functional changes (e.g., defective platelet granule secretion), or platelet-wall interaction abnormalities can cause bleeding diathesis in patients undergoing HD. Anemia beyond platelet abnormalities can be associated with bleeding complications [21,22]. Therefore, patients undergoing HD and administered warfarin are exposed to a high bleeding risk. 

Previous clinical studies have reported abnormalities in platelet function in patients undergoing dialysis, and these changes are ultimately associated with excessive clotting and bleeding [23,24,25,26]. The presence of platelet dysfunction may have influenced the results of our study. However, despite the significance of platelet dysfunction in patients undergoing HD, its measurement methods and clinical significance remain unclear. These changes in platelet function arise from various abnormalities in platelet morphology, function, and interactions with the vessel wall, and their clinical manifestations are diverse, necessitating specific identification [26]. Conventionally, platelet function can be assessed using distinct data such as skin bleeding time, closure time, and traditional aggregometry [26]. However, these methods lack sufficient validation in patients undergoing dialysis, exhibit poor inter-test correlation, and have inconsistent clinical significance [27,28]. Consequently, data analysis on platelet function is not commonly performed clinically, and the dataset used in our study, which only collected minimal data for HD quality assessment, does not include information on this. Recent research has explored noncomplex approaches using platelet volume and count to assess thrombosis and bleeding risk; however, further studies are needed [29]. 

Non-vitamin K oral anticoagulants, including apixaban, dabigatran, edoxaban, and rivaroxaban, are used for the prevention of thromboembolic disease; these medications do not require regular drug monitoring and have a more rapid onset of action, shorter half-life, and fewer drug interactions than warfarin [30]. However, dabigatran, edoxaban, and rivaroxaban are contraindicated in patients undergoing dialysis. Furthermore, compared to no medications or warfarin use, apixaban use has insufficient evidence in terms of safety or complications, despite FDA approval for use in patients undergoing HD. Recent randomized trials have compared non-vitamin K antagonists and warfarin in patients with atrial fibrillation who were undergoing HD. De Vriese et al. compared warfarin and low-dose rivaroxaban and revealed a decreased risk of fatal and non-fatal CVE and bleeding complications with low-dose rivaroxaban but similar all-cause mortality [31]. Two randomized studies compared the efficacy and complications between apixaban and warfarin therapies in patients with atrial fibrillation who were undergoing HD and found no statistical differences in safety and efficacy [32,33]. 

Randomized trials have compared the administration of two anticoagulants in patients with atrial fibrillation who were undergoing HD. However, most studies comparing the use and non-use of warfarin have involved retrospective or non-controlled prospective cohort designs. Shen et al. analyzed the clinical outcomes of 12,284 patients who were administered warfarin after the first diagnosis of atrial fibrillation [34]. Their study revealed a minimal decrease in the ischemic stroke risk and a similar decline in mortality, bleeding complications, and hemorrhagic stroke outcomes, thereby emphasizing the minimal benefits of long-term use of warfarin. However, other studies comparing warfarin use to non-use showed the hazardous effects of warfarin. Sy et al. showed higher risks of bleeding and hemorrhagic stroke after considering death as a competing risk [12]. Yoon et al. evaluated 9974 patients with atrial fibrillation who were undergoing HD using a population-based cohort and revealed a neutral effect on ischemic stroke and a hazardous effect on hemorrhagic stroke when comparing warfarin-use cases with those of non-use [11]. 

Our study had certain strengths compared to previous studies, including a relatively large sample size and a long follow-up period. A meta-analysis of 15 studies involved a median follow-up of 2.6 years, but our study had a mean follow-up of 53 months in the No group and 50 months in the Warfarin group [6]. Additionally, our study included laboratory data associated with HD quality. Two previous studies used datasets similar to those used in our study [11,13]. Yoon et al. evaluated a larger sample size than that in our study, but they did not include laboratory or mortality data [11]. Kang et al. included laboratory data; however, our study included a larger sample size than that in their study [13]. 

The results of subgroup analyses in our study provided additional information. In female patients, the use of warfarin was associated with decreased CVE, increased all-cause mortality, and a neutral effect on hemorrhagic stroke. In multivariable Cox regression, the nonsignificant relationship between hemorrhagic stroke risk and the use of warfarin in female patients may be linked to the small sample size of female warfarin users (n = 219) and low incidence of hemorrhagic stroke. Considering the high risk of hemorrhagic stroke in most subgroups, high all-cause mortality in females or young patients, and its neutral effect on ischemic stroke or CVE, warfarin should be used with caution. The use of warfarin in patients with a CHA2DS2-VASc score ≥ 3 in females or ≥ 2 in males is strongly recommended [9,10]. However, in our study, the use of warfarin in patients with these scores exhibited a trend towards higher all-cause mortality, despite statistical non-significance, and a significantly increased risk of hemorrhagic stroke. This suggests that caution should be exercised while using warfarin, even for individuals with high scores in the general population.

Our study aimed to investigate the association between warfarin use and clinical outcomes in patients with atrial fibrillation who were undergoing HD and identify factors influencing the risk of all-cause mortality and hemorrhagic stroke in these patients. In these patients, age, the presence of diabetes, serum calcium and phosphorus levels, HD vintage, CCI scores, UFV, the use of clopidogrel, female sex, BMI, Kt/V_urea_, and levels of hemoglobin, albumin, and creatinine played a role in all-cause mortality. However, the factors influencing the risk of death in our study were similar to well-known mortality risk factors in patients undergoing HD, regardless of the presence of atrial fibrillation. Hemorrhagic stroke was positively associated with calcium levels, which aligns with the findings of Kitamura et al., who demonstrated an association between serum calcium levels and cerebral hemorrhage [35]. While evidence regarding the association between serum calcium levels and hemorrhagic stroke is insufficient, it may be related to vascular calcification, changes in the vascular tone, and the function of the blood–brain barrier [36,37,38,39,40]. Further research is needed to elucidate the mechanisms of and establish a clear association between serum calcium levels and hemorrhagic stroke.

Our study had certain limitations. First, our study had a retrospective observational design. The initial collection of our study data was for quality assessments at the HD center, independent of research objectives. Nonetheless, our dataset represents real-world scenarios compared to data obtained for research purposes in single-center or multicenter studies. Second, warfarin use was defined in our study as its utilization over a specified period during the HD quality assessment program (i.e., used ≥ 30 days in 6 months). Third, we evaluated the diagnosis of atrial fibrillation, stroke, or CVE using ICD-10 codes or procedures. Atrial fibrillation can be precisely diagnosed using electrocardiography, 24 h Holter monitoring, or echocardiography. In addition, our study did not include structural or functional changes in the heart. Ejection fraction, wall motion, or valve abnormality data would have been useful in identifying the types or underlying comorbidities associated with atrial fibrillation. Fourth, data on the cause of death were not included in our study analysis. Most of the South Korean population is enrolled in medical insurance, or Medicaid, and indicating the status of treatment-related mortality on insurance claims is mandatory. Therefore, the mortality data provided by the HIRA is highly accurate. The mortality status is reported to be verified by the continued payment of insurance premiums; thus, information regarding the cause of death is not collected. As a result, our dataset does not allow for data analysis related to the cause of death.

Our study is retrospective and is not confirmative due to several inherent limitations. However, we believe it is valuable as it analyzes real-world data on the use of warfarin in HD patients with atrial fibrillation, for which clear conclusions are yet to be drawn. Moreover, we believe that this study highlights the need for further confirmative randomized controlled trials based on our findings.

## 5. Conclusions

We showed that warfarin use is associated with a higher risk of all-cause mortality and hemorrhagic stroke and shows a neutral effect on ischemic stroke and CVE in patients with atrial fibrillation who were undergoing HD. These findings are consistent across age, sex, diabetes, and CHA_2_DS_2_-VAS_C_ score subgroups. These results underscore the importance of exercising caution in the use of warfarin, irrespective of the thromboembolic risk or baseline characteristics, in patients with atrial fibrillation who are undergoing HD.

## Figures and Tables

**Figure 1 jcm-13-02404-f001:**
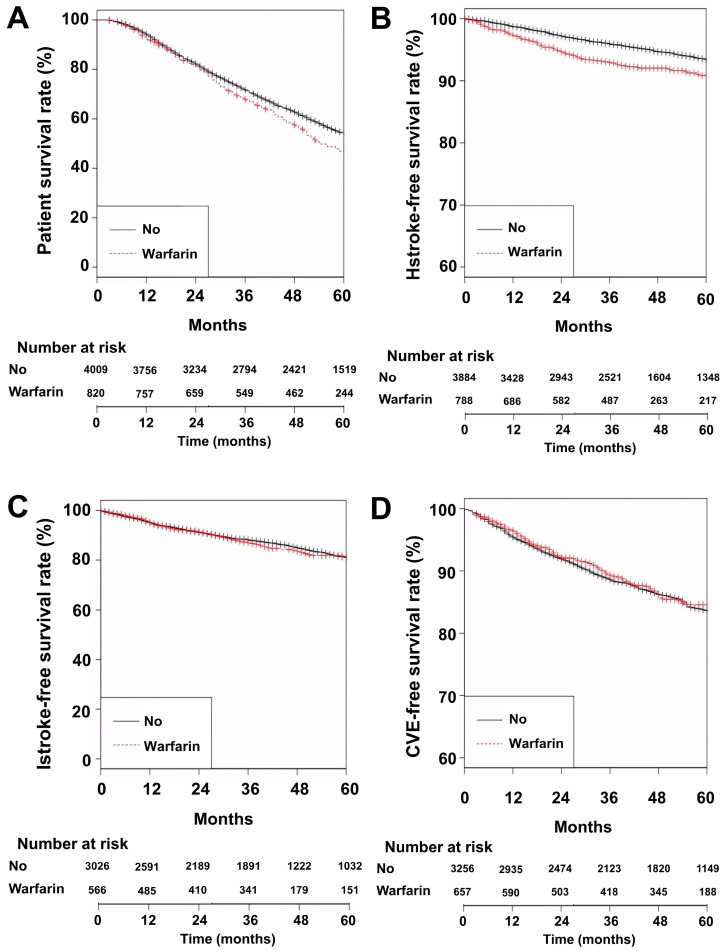
Kaplan–Meier curves for all-cause mortality, hemorrhagic stroke, ischemic stroke, and cardiovascular events according to the use of warfarin. (**A**) All-cause mortality; (**B**) Hemorrhagic stroke; (**C**) Ischemic stroke; (**D**) Cardiovascular events. Abbreviations: CVE, cardiovascular events; Hstroke, hemorrhagic stroke; Istroke, ischemic stroke.

**Figure 2 jcm-13-02404-f002:**
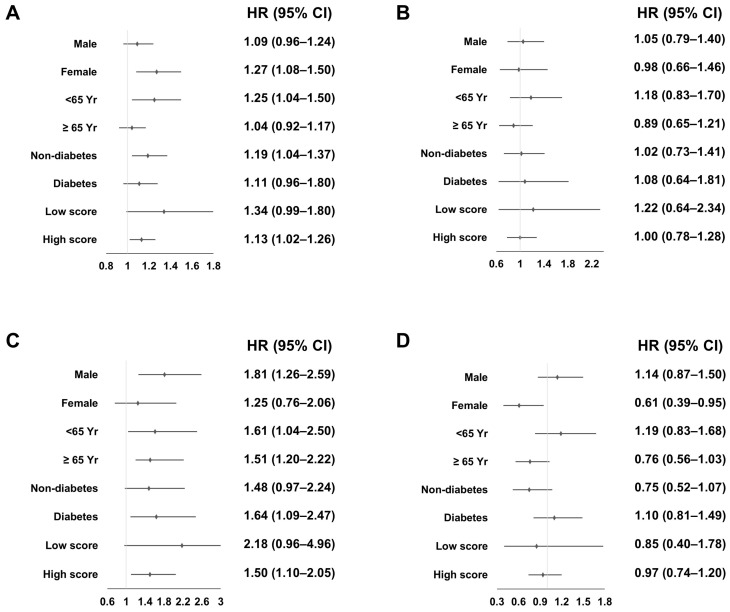
Association between the use of warfarin and clinical outcomes using univariate Cox regression analyses based on subgroups. (**A**) All-cause mortality; (**B**) Ischemic stroke; (**C**) Hemorrhagic stroke; (**D**) Cardiovascular events. A low CHA_2_DS_2_-VAS_C_ score was <2 in females and <3 in males, and a high CHA_2_DS_2_-VAS_C_ score was ≥2 in females and ≥3 in males. The values represent the HR of the Warfarin group compared to the No group. Abbreviations: CI, confidence interval; HR, hazard ratio.

**Figure 3 jcm-13-02404-f003:**
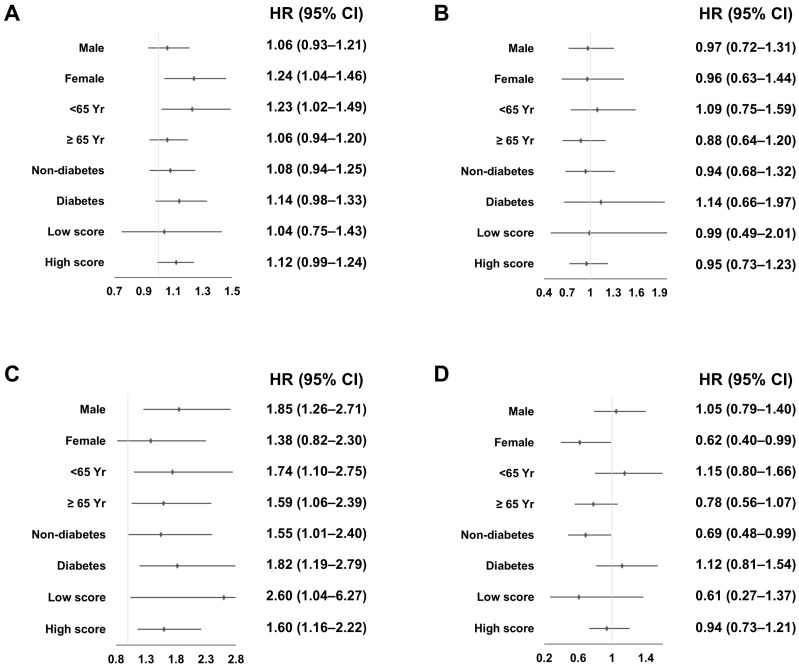
Association between the use of warfarin and clinical outcomes using multivariable Cox regression analyses based on subgroups. (**A**) All-cause mortality; (**B**) Ischemic stroke; (**C**) Hemorrhagic stroke; (**D**) Cardiovascular events. A low CHA_2_DS_2_-VAS_C_ score was <2 in females and <3 in males, and a high CHA_2_DS_2_-VAS_C_ score was ≥2 in females and ≥3 in males. The values represent the HR of the Warfarin group compared to the No group. Abbreviations: CI, confidence interval; HR, hazard ratio.

**Table 1 jcm-13-02404-t001:** Patient baseline characteristics.

	No Group (n = 4009)	Warfarin Group (n = 820)	*p*-Value
Age (years)	65.0 ± 11.6	66.9 ± 10.9	<0.001
Sex (male, %)	2471 (61.6%)	527 (64.3%)	0.169
Hemodialysis vintage (days)	57 ± 62	53 ± 59	0.091
Body mass index (kg/m^2^)	22.4 ± 3.5	22.4 ± 3.4	0.760
Underlying cause of ESKD (diabetes)	1714 (42.8%)	363 (44.3%)	0.448
CCI score	8.6 ± 2.9	8.8 ± 2.7	0.254
Kt/V_urea_	1.52 ± 0.25	1.51 ± 0.25	0.262
UFV (L/session)	2.3 ± 0.9	2.3 ± 0.8	0.688
Hemoglobin (g/dL)	10.6 ± 0.9	10.7 ± 0.9	0.018
Serum albumin (g/dL)	3.93 ± 0.35	3.91 ± 0.33	0.088
Serum phosphorus (mg/dL)	4.8 ± 1.3	4.7 ± 1.3	0.026
Serum calcium (mg/dL)	8.9 ± 0.8	8.9 ± 0.8	0.421
Serum creatinine (mg/dL)	8.9 ± 2.6	8.7 ± 2.5	<0.001
Use of aspirin	2378 (59.3%)	319 (38.9%)	<0.001
Use of clopidogrel	1268 (31.6%)	138 (16.8%)	<0.001
Use of statins	1681 (41.9%)	396 (48.3%)	<0.001
Use of β-blockers	2070 (51.6%)	450 (54.9%)	0.098
CHA_2_DS_2_-VAS_C_ score	3.4 ± 1.6	3.4 ± 1.6	0.705
HAS-BLED score	5.1 ± 1.1	4.9 ± 1.1	<0.001

Mean ± standard deviation is used to represent continuous variables, while categorical variables are presented as numbers (percentages). The *p*-values are determined using a *t*-test, and Pearson’s χ^2^ test was performed for categorical variables. Abbreviations: CCI, Charlson Comorbidity index; ESKD, end-stage kidney disease; UFV, ultrafiltration volume.

**Table 2 jcm-13-02404-t002:** Cox regression analyses for patient survival.

	Univariate		Multivariable	
	HR (95% CI)	*p*	HR (95% CI)	*p*
All-cause mortality	1.15 (1.05–1.28)	0.005	1.11 (1.01–1.23)	0.047
Ischemic stroke	1.02 (0.81–1.29)	0.847	0.96 (0.76–1.22)	0.744
Hemorrhagic stroke	1.57 (1.17–2.10)	0.002	1.67 (1.23–2.27)	<0.001
Cardiovascular events	0.93 (0.74–1.18)	0.568	0.91 (0.71–1.15)	0.431

The values represent the HR of the Warfarin group compared to the No group. Adjustments in multivariable analysis included body mass index, vascular access type, age, sex, diabetes, Charlson Comorbidity Index score, hemodialysis vintage, ultrafiltration volume, Kt/V_urea_, serum albumin, hemoglobin, serum calcium, serum creatinine, serum phosphorus, systolic blood pressure, diastolic blood pressure, and use of antihypertensive drug, aspirin, and statins. The analysis was conducted using the enter mode. Abbreviations: CI, confidence interval; HR, hazard ratio.

## Data Availability

The raw data were generated at the Health Insurance Review and Assessment Service. The database can be requested from the Health Insurance Review and Assessment Service by sending a study proposal including the purpose of the study, study design, and duration of analysis through an e-mail (turtle52@hira.or.kr) or at the website (https://www.hira.or.kr (accessed on 18 April 2024)). The authors cannot distribute the data without permission.

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
