# Peer review of "Clinical Impact of the Use of Warfarin in Patients with Atrial Fibrillation Undergoing Maintenance Hemodialysis"

_jcm, 2024, doi:10.3390/jcm13082404_

Round 1

Reviewer 1 Report

Comments and Suggestions for Authors

The use of warfarin in patients with chronic kidney disease (CKD) on hemodialysis (HD) and atrial fibrillation (AF) is very questionable. However, studies to answer this question are limited. A retrospective analysis of this type of patient undergoing qualified HD programs was performed to evaluate the benefit of warfarin compared to those without warfarin over an average period of 4 years. Warfarin in these patients had no benefit, on the contrary, a higher risk of all-cause mortality and hemorrhagic stroke. In this series, independent risk factors for hemorrhagic stroke were evaluated in patients with diabetes and higher levels of plasma calcium and a lower risk in patients with increased body mass index. The importance of this study with an adequate sample and longer evolution time undeniably suggests that warfarin does not bring benefits but rather harms in patients with CKD undergoing HD associated with the presence of AF.

Author Response

Answer: Thank you for your valuable comments. Our study is a retrospective study and is not confirmative due to several inherent limitations. However, we believe it is valuable as it analyzes real-world data on the use of warfarin in HD patients with atrial fibrillation, for which clear conclusions are yet to be drawn. Additionally, we believe that this study highlights the need for further confirmative randomized controlled trials based on our findings. We have added these comments in the Discussion section.

Reviewer 2 Report

Comments and Suggestions for Authors

Dear Dr.,

Title: Clinical impact of the use of warfarin in patients with atrial fibrillation undergoing maintenance hemodialysis

Manuscript ID: jcm-2958541

Overall comments: Kang et al. described in this manuscript: the warfarin uses for the maintenance of in hemodialysis patients with atrial fibrillation. The major limitation, the clarity of warfarin associated other complications can be discussed. The overall manuscript is good and it can help those working in this field of research.

Specific comments:

1.      The abstract is written well.

2.      In this manuscript can be highlight the inclusion and exclusion criteria for the assessment of this hypothesis.

3.      Figure 1. Is blurred, need to make clear image.

4.      Discussion is too lengthy.

5.      Future perspectives can be incorporate in the discussion or conclusion sections.

Minor comments

1.      Recent references can be incorporated to relevant statements.

*****

Author Response

Overall comments: Kang et al. described in this manuscript: the warfarin uses for the maintenance of in hemodialysis patients with atrial fibrillation. The major limitation, the clarity of warfarin associated other complications can be discussed. The overall manuscript is good and it can help those working in this field of research.

Specific comments:

  1. The abstract is written well.
  2. In this manuscript can be highlight the inclusion and exclusion criteria for the assessment of this hypothesis.

Answer: Thank you for your comment. To visually represent the inclusion and exclusion criteria, we have added Figure S1 as a study flow chart.

Figure S1. Study flow chart

Abbreviations: HD, hemodialysis; ICD-10, International Classification of Disease-10th version.

  1. Figure 1. Is blurred, need to make clear image.

Answer: Thank you for your comment. We have revised “Figure 1 with 300 dpi” to “Figure 1 with 600dpi”.

  1. Discussion is too lengthy.

Answer: Thank you for your comment. However, the journal requires minimum word count of 4000 words in the main text. In our opinion, a long Discussion section would make reviewers and audiences better understand the research background, perspectives, and novelty of our study.

  1. Future perspectives can be incorporate in the discussion or conclusion sections.

Answer: Thank you for your comment.

Our study is a retrospective study and is not confirmative due to several inherent limitations. However, we believe it is valuable as it analyzes real-world data on the use of warfarin in HD patients with atrial fibrillation, for which clear conclusions are yet to be drawn. Additionally, we believe that this study highlights the need for further confirmative randomized controlled trials based on our findings.

We have added these future perspectives in the Discussion section.

Minor comments

  1. Recent references can be incorporated to relevant statements.

Answer: Thank you for your comment. We have added the following recent references to the study:

van Eck van der Sluijs A, Pai P, Zhu W, Ocak G. Bleeding Risk in Hemodialysis Patients. Semin Nephrol. 2024 Jan 17:151478.

Warkentin TE. Immunologic Effects of Heparin Associated With Hemodialysis: Focus on Heparin-Induced Thrombocytopenia. Semin Nephrol. 2024 Jan 8:151479. 

Gomchok D, Ge RL, Wuren T. Platelets in Renal Disease. Int J Mol Sci. 2023 Sep 29;24(19):14724.

Wang J, Li W, Zhang W, Cao L. Association between serum calcium and hemorrhagic transformation in ischemic stroke: A systematic review and meta-analysis. J Clin Neurosci. 2022 Oct;104:107-112.

Bkaily G, Jacques D. Calcium Homeostasis, Transporters, and Blockers in Health and Diseases of the Cardiovascular System. Int J Mol Sci. 2023 May 15;24(10):8803. 

Reviewer 3 Report

Comments and Suggestions for Authors

The study conducted by Seok-Hui Kang et al, entitled: "Clinical impact of the use of warfarin in patients with atrial fibrillation undergoing maintenance hemodialysis" share light on the dilemma of Warfarin use in patients with atrial fibrillation who need to hemodialysis.

Overall, the paper is well written. The authors have done a great job to a certain extend by providing subgroups analysis which is one of the strengths of this study.

However, some sections of the manuscript need to be double checked for proofreading and language style.

Comments on the Quality of English Language

The English level used in this manuscript is overall good, even though some parts need to be re-checked.

Author Response

The study conducted by Seok-Hui Kang et al, entitled: "Clinical impact of the use of warfarin in patients with atrial fibrillation undergoing maintenance hemodialysis" share light on the dilemma of Warfarin use in patients with atrial fibrillation who need to hemodialysis.

Overall, the paper is well written. The authors have done a great job to a certain extend by providing subgroups analysis which is one of the strengths of this study.

However, some sections of the manuscript need to be double checked for proofreading and language style.

Answer: Thank you for your valuable comments. As suggested by the reviewer, we have rechecked the paper for English language.

Reviewer 4 Report

Comments and Suggestions for Authors

Dear authors 

thank you for this effort 

these some notes 

Line

Present 

Comment 

40

impaired platelet function

Impaired platelet function is important issue, involving bleeding time in this study 

Table no 1 

P value of age is less than 0.001 

Demographic parameter at base line study, better to be identical 

Table no 1 

Arteriovenous fistula p value 1 

Better to remove from table 

Table no 1 

Serum phosphorus p value less than 0.05 

The two group are not identical, may be interfere with the results 

Table no 1 

Serum calcium  p value less than 0.05 

The two group are not identical, may be interfere with the results 

Table no 1

Use of aspirin and other drugs 

The two group are not identical, may be interfere with the results 

Figure one

Kaplan Meier

No censer cases ????

226 

Effect of platelet dysfunction 

Better to be included in this study 

Author Response

Line 40: impaired platelet function; Impaired platelet function is important issue, involving bleeding time in this study

Answer: Thank you for your comments.

Previous clinical studies have reported abnormalities in platelet function in patients undergoing dialysis, and these changes are ultimately associated with excessive clotting and bleeding [1-4]. The presence of platelet dysfunction may have influenced the results of our study. However, despite the significance of platelet dysfunction in patients undergoing HD, its measurement methods and clinical significance are not completely understood. These changes in platelet function arise from various abnormalities in platelet morphology, function, and interactions with the vessel wall, and their clinical manifestations are diverse, necessitating specific identification [4]. Classically, assessment of platelet function can be done using special data such as skin bleeding time, closure time, and traditional aggregometry [4]. However, these methods lack sufficient validation in patients undergoing dialysis, exhibit poor inter-test correlation, and have inconsistent clinical significance [5,6]. Consequently, data analysis on platelet function is not commonly performed clinically, and the dataset used in our study, which only collected minimal data for HD quality assessment, does not include information on this. Recent research has explored simpler approaches using platelet volume and count to assess thrombosis and bleeding risk; however, further studies are needed [7].

We have added these comments in the Discussion section.

Added references

[1] Remuzzi G, Livio M, Marchiaro G, Mecca G, de Gaetano G. Bleeding in renal failure: altered platelet function in chronic uraemia only partially corrected by haemodialysis. Nephron. 1978;22(4-6):347-53.

[2] Bazilinski N, Shaykh M, Dunea G, Mamdani B, Patel A, Czapek E, Ahmed S. Inhibition of platelet function by uremic middle molecules. Nephron. 1985;40(4):423-8.

[3] Di Minno G, Martinez J, McKean ML, De La Rosa J, Burke JF, Murphy S. Platelet dysfunction in uremia. Multifaceted defect partially corrected by dialysis. Am J Med. 1985 Nov;79(5):552-9.

[4] Jain N, Corken AL, Kumar A, Davis CL, Ware J, Arthur JM. Role of Platelets in Chronic Kidney Disease. J Am Soc Nephrol. 2021;32(7):1551-1558.

[5] Zupan IP, Sabovic M, Salobir B, Ponikvar JB, Cernelc P. Utility of in vitro closure time test for evaluating platelet-related primary hemostasis in dialysis patients. Am J Kidney Dis. 2003 Oct;42(4):746-51.

[6] Wright RS, Anderson JL, Adams CD, Bridges CR, Casey DE Jr, Ettinger SM, Fesmire FM, Ganiats TG, Jneid H, Lincoff AM, Peterson ED, Philippides GJ, Theroux P, Wenger NK, Zidar JP, Jacobs AK. 2011 ACCF/AHA Focused Update of the Guidelines for the Management of Patients With Unstable Angina/ Non-ST-Elevation Myocardial Infarction (Updating the 2007 Guideline): a report of the American College of Cardiology Foundation/American Heart Association Task Force on Practice Guidelines. Circulation. 2011 May 10;123(18):2022-60.

[7] Davis OM, Kore R, Moore A, Ware J, Mehta JL, Arthur JM, Lynch DR, Jain N. Platelet Count and Platelet Volume in Patients with CKD. J Am Soc Nephrol. 2023 Nov 1;34(11):1772-1775.

Table no 1: P value of age is less than 0.001; Demographic parameter at base line study, better to be identical 

Table no 1: Serum phosphorus p value less than 0.05; The two group are not identical, may be interfere with the results 

Table no 1: Serum calcium  p value less than 0.05 ; The two group are not identical, may be interfere with the results 

Table no 1: Use of aspirin and other drugs; The two group are not identical, may be interfere with the results.

Answer: Thank you for your comments. We have performed analyses using cohort after propensity score matching. Distribution of propensity score before and after matching are presented in Figure S2. Baseline characteristics after propensity score matching are shown in Table S4; differences in most baseline characteristics were attenuated after matching. Cox regression analyses using cohort after propensity score matching are presented in Table S5. Results using cohort after propensity score matching were similar with those using total cohort.

Figure S2. Distribution of propensity score before and after matching.

Table S4. Baseline characteristics after propensity score matching

No group

(n = 1,621)

Warfarin group

(n = 820)

P-value

Age (years)

67.0 ± 11.4

66.9 ± 10.9

0.888

Sex (male, %)

1015 (62.6%)

527 (64.3%)

0.450

Hemodialysis vintage (days)

52 ± 58

53 ± 59

0.812

Body mass index (kg/m2)

22.4 ± 3.6

22.4 ± 3.4

0.934

Underlying cause of ESKD (diabetes)

919 (56.7%)

363 (44.3%)

0.682

CCI score

8.8 ± 2.9

8.8 ± 2.7

0.942

Kt/Vurea

1.52 ± 0.24

1.51 ± 0.25

0.732

UFV (L/session)

2.3 ± 0.9

2.3 ± 0.8

0.890

Hemoglobin (g/dL)

10.7 ± 0.9

10.7 ± 0.9

0.536

Serum albumin (g/dL)

3.92 ± 0.35

3.91 ± 0.33

0.504

Serum phosphorus (mg/dL)

4.7 ± 1.3

4.7 ± 1.3

0.946

Serum calcium (mg/dL)

8.9 ± 0.8

8.9 ± 0.8

0.400

Serum creatinine (mg/dL)

8.6 ± 2.5

8.7 ± 2.5

0.887

Use of aspirin

627 (38.7%)

319 (38.9%)

0.950

Use of clopidogrel

298 (18.4%)

138 (16.8%)

0.373

Use of statins

602 (37.1%)

396 (48.3%)

<0.001

Use of β-blockers

858 (52.9%)

450 (54.9%)

0.385

Data are expressed as mean ± standard deviation for continuous variables and as numbers (percentages) for categorical variables. P-values are determined using a t-test, and Pearson’s χ2 test was performed for categorical variables. Abbreviations: CCI, Charlson Comorbidity index; ESKD, end-stage kidney disease; UFV, ultrafiltration volume

Table S5. Cox regression analyses using cohort after propensity score matching

Univariate

Multivariable

HR (95% CI)

P

HR (95% CI)

P

All-cause mortality

1.15 (1.05–1.28)

0.005

1.11 (1.01–1.23)

0.047

Ischemic stroke

1.02 (0.81–1.29)

0.847

0.96 (0.76–1.22)

0.744

Hemorrhagic stroke

1.57 (1.17–2.10)

0.002

1.67 (1.23–2.27)

<0.001

Cardiovascular events

0.93 (0.74–1.18)

0.568

0.91 (0.71–1.15)

0.431

Adjustments in multivariable analysis included body mass index, vascular access type, age, sex, diabetes, Charlson Comorbidity Index score, hemodialysis vintage, ultrafiltration volume, Kt/Vurea, serum albumin, hemoglobin, serum calcium, serum creatinine, serum phosphorus, systolic blood pressure, diastolic blood pressure, and use of anti–hypertensive drug, aspirin, and statins. The analysis was conducted using the enter mode. Abbreviations: CI, confidence interval; HR, hazard ratio.

Table no 1: Arteriovenous fistula p value 1; Better to remove from table 

Thank you for your comment. We have deleted the comment for vascular access in Table 1 and have added a comment in the main body.

Figure one: Kaplan Meier; No censer cases ????

Answer: Thank you for your comment. We have revised the number of people at risk and the censor points presented in Figure 1 at bottom. The revised version of Figure 1 is as follows:

Line 226: Effect of platelet dysfunction; Better to be included in this study 

Answer: Thank you for your comments. Our study did not include data for platelet dysfunction. We have added some comments regarding platelet dysfunction in patients undergoing HD. Detailed explanations have been provided.

Reviewer 5 Report

Comments and Suggestions for Authors

This study is a retrospective analysis to evaluate the use of warfarin in patients receiving maintenance therapy for atrial fibrillation (AF) on hemodialysis (HD). The following key points and my review of the potential are suitable for publication:
Aims Clarity: Objectives for assessing the clinical impact of warfarin use The outcome for patients with atrial fibrillation receiving Huntington's disease is clear. This is related to addressing an important clinical question.
METHODOLOGY METHODS: The study utilized a retrospective analysis of adult data on patients receiving maintenance therapy for Huntington's disease. The use of a large laboratory queue of clinical findings related to HD quality added to the strength of the study.
Sample Size and Duration: Enrollment of 4,829 patients, with a significant subgroup of 820 treated with warfarin, suggests a large dataset. The duration of follow-up was reasonable.
Statistical analysis: Cox regression analysis was used to assess the association between warfarin use and clinical outcomes. The reported risk ratios for all-cause mortality and hemorrhagic stroke provide quantitative insight into the observed associations.
Key Findings: Findings suggest that warfarin use in AF patients receiving Huntington's chorea is associated with a higher risk of all-cause mortality and hemorrhagic stroke. There was no significant association with ischemic stroke or cardiovascular events of note. These results provide valuable insight into the clinical management of patients with AF who receive Huntington's disease.
Overall The study appears to address an important clinical question using robust methods and a large dataset.

Author Response

(The authors gave the same response as above.)
